# An Investigation into the Effect of Maltitol, Sorbitol, and Xylitol on the Formation of Carbamazepine Solid Dispersions Through Thermal Processing

**DOI:** 10.3390/pharmaceutics17030321

**Published:** 2025-03-02

**Authors:** Madan Sai Poka, Marnus Milne, Anita Wessels, Marique Aucamp

**Affiliations:** 1Department of Pharmaceutical Sciences, School of Pharmacy, Sefako Makgatho Health Sciences University, Pretoria 0208, South Africa; marnus.milne@smu.ac.za; 2Centre of Excellence for Pharmaceutical Sciences (Pharmacen), North-West University, Private Bag X6001, Potchefstroom 2520, South Africa; anita.wessels@nwu.ac.za; 3School of Pharmacy, University of the Western Cape, Robert Sobukwe Drive, Bellville, Cape Town 7130, South Africa; maucamp@uwc.ac.za

**Keywords:** carbamazepine, polyols, solid dispersions, polymorphic forms, aqueous solubility, dissolution enhancement

## Abstract

**Background:** Carbamazepine (CBZ) is a Biopharmaceutical Classification System (BCS) class II drug, that is practically insoluble in water, influencing the oral bioavailability. Polyols are highly hydrophilic crystalline carriers studied for their success in developing solid dispersions (SDs) for improved solubility and dissolution rate. Polyols are generally regarded as safe (GRAS) and maltitol (MAL), xylitol (XYL) and sorbitol (SOR) are among the approved polyols for market use. While xylitol (XYL) and sorbitol, have shown promise in improving the solubility and dissolution rates of poorly soluble drugs, their full potential in the context of improving the solubility of carbamazepine have not been thoroughly investigated. To the best of our knowledge, maltitol (MAL) was not studied previously as a carrier for preparing SDs. Hence, the purpose of this study was to investigate their use in the preparation of CBZ SDs by the fusion method. **Methods:** CBZ-polyol SDs were prepared in varying molar ratios (2:1, 1:1 and 1:2) and characterised for solid-state nature, solubility and in-vitro dissolution rate. **Results:** Solid-state characterisation of the CBZ-polyol SDs revealed the existence of the SDs as continuous glass suspensions with fine CBZ crystallites suspended in the amorphous polyol carriers. Among the polyols studied, XYL exhibited good miscibility with CBZ and showed significant improvement in the solubility and dissolution rate. The prepared SDs showed a 2 to 6-folds increase in CBZ solubility and 1.4 to 1.9-folds increase in dissolution rate in comparison with pure CBZ. **Conclusions:** The study explains the possible use of polyols (XYL and SOR) based SDs of BCS Class II drugs with good glass forming ability for enhanced solubility and dissolution.

## 1. Introduction

Carbamazepine (CBZ) is an important first-line antiepileptic drug classified as a Biopharmaceutical Classification System (BCS) class II drug due to its poor aqueous solubility of less than 200 µg/mL [1,2]. The commercially available CBZ polymorph Form III is commonly used in the formulation of oral drug delivery systems due to its thermodynamic stability [3]. However, its poor aqueous solubility results in the slow and irregular oral absorption of CBZ, causing either therapeutic failure or increased toxicity [4].

Since the introduction of the solid dispersion (SD) preparation technique in the early 1960s [5], a plethora of studies have been conducted and reported for their success in enhancing the solubility and dissolution of CBZ. Among the various studies conducted, the majority of them reported the use of hydrophilic carriers such as sugars [6,7,8], polyethylene glycols (PEGs) [9,10,11,12], and polymers such as polyvinylpyrrolidone [2,13,14], co-polyvidone (Kollidon VA64) [14,15], Soluplus^®^ [15,16], hydroxy propyl methyl cellulose [14,17], etc. These carriers come with unique challenges such as the low degradation threshold of sugars [18], poor glass-forming ability, and great variation in the physicochemical properties of PEGs as a function of processing parameters during manufacturing, thereby limiting their scalability [5]. In the case of polymers, hygroscopicity, high melting point, and high viscosity in the solution are some of the major challenges reported to have negative impacts on the drug loading and stability of the SDs [19]. Considering this, there is a need for continuous research to identify new carriers that can address some of these challenges and most importantly show the potential application in commercial manufacturing.

Polyols, also known as polyhydric alcohols or sugar alcohols, have been long used for pharmaceutical purposes due to their strong inertness and well-known safety in terms of human consumption [20,21]. Polyols such as mannitol, maltitol, sorbitol, and xylitol are extensively used as sweeteners in orally disintegrating tablets [20,22], bulk-forming agents in freeze-dried products [23], diluents, plasticizers, humectants, and disintegrant in the formulation of tablets and capsules [24,25]. Several studies reported the use of polyols as potential carriers in the preparation of SDs for the solubility and dissolution enhancement of poorly soluble drugs. Amongst the United States Food and Drug Administration (FDA)-approved polyols [26], mannitol has been extensively studied in comparison with other polyols such as xylitol, sorbitol, erythritol, and maltitol. Recently, Narala et al. [27] reported the use of xylitol as a promising carrier for the solubility and dissolution enhancement of CBZ. The study reported successfully preparing a CBZ solid crystalline suspension using hot melt extrusion resulting in a 50-fold increase in the aqueous solubility of CBZ and more than 85% drug release in 15 min. The prepared SDs showed good stability for a period of 3 months at accelerated storage conditions. Given the promising results, the increasing application of polyols in the pharmaceutical manufacturing sector, taste-masking capabilities, and relatively low cost make them interesting candidates to be used as carriers in the preparation of SDs. Thus, the current study aims at investigating the selected polyols (maltitol, sorbitol, and xylitol) in the preparation of CBZ-polyol SDs using the well-known fusion method and their effect on solubility and dissolution enhancement.

## 2. Materials and Methods

### 2.1. Materials

Carbamazepine (CBZ; ≥97%), Figure 1a was purchased from Glentham Life Sciences Ltd. (Corsham, UK). Maltitol (MAL; ≥98.0%), D-sorbitol (SOR; ≥97%), and xylitol (XYL; ≥98.5%), Figure 1b–d, was purchased from Glentham Life Sciences Ltd. (Corsham, UK). Chromatography-grade methanol was obtained from Labchem (Johannesburg, South Africa) and ultrapure water with a resistivity of 18.2 MΩ.cm was obtained from an Elga Veolia (High Wycombe, UK) water purification system. For the preparation of aqueous solutions, deionized water was filtered by a Nanopure™ Water Purification System (Thermo Scientific, Waltham, MA, USA). All the materials were used as provided and no further purification process was performed.

### 2.2. Methods

#### 2.2.1. Preparation of Physical Mixtures

The binary physical mixtures (PMs) of CBZ-MAL, CBZ-SOR, and CBZ-XYL were prepared by gently grinding the accurately weighed quantities of CBZ, MAL, SOR, and XYL at the different drug to sugar molar weight ratios of 2:1, 1:1, and 1:2. In each instance, CBZ and the corresponding polyol were geometrically mixed using a mortar and pestle for 2 min to prepare a homogenous mixture. The physical mixtures were subsequently stored in a desiccator until further use.

#### 2.2.2. Preparation of Solid Dispersions

Solid dispersions (SDs) of CBZ-MAL, CBZ-SOR, and CBZ-XYL were prepared by the well-known fusion method [1]. Physical mixtures of the different drug to polyol ratios were placed in a porcelain dish and melted on a heating mantle (Heidolph, Germany) at 210 ± 5 °C. The molten product was subsequently cooled to room temperature (RT). RT cooling was performed by simply leaving the molten product at room temperature (25 °C) until it was solidified. The solidified material was then stored in desiccators for 24 h, before pulverisation using a mortar pestle.

#### 2.2.3. Physicochemical Characterisation of Prepared Solid Dispersions

##### Differential Scanning Calorimetry (DSC)

CBZ, MAL, SORB, XYL, PMs, and the prepared SDs were analysed using a Mettler Toledo DSC3, (Columbus, OH, USA). Approximately 5–10 mg of each sample was weighed into aluminium pans which were subsequently sealed with a pin-holed aluminium lid. The samples were heated from 30 to 300 °C at a constant heating rate of 10 °C/min under a continuous flow of nitrogen at 50 mL/min. Analysis of the data was performed using the STARe software, version 17.00, (Mettler Toledo, Columbus, OH, USA).

##### Hot-Stage Microscopy (HSM)

Analysis of the pure compounds as well as PMs and SDs was performed using a real-time Olympus UC30 (Tokyo, Japan) camera fitted to an Olympus SZX-ILLB200 (Tokyo, Japan) polarised light microscope to which a Linkam THMS600 heating stage equipped with a T95 LinkPad temperature controller (Surrey, UK) was attached. A small quantity of each sample was placed in between two microscope glass slides and heated at a heating rate of 10 °C/min. Photomicrographs were acquired at 40× magnification and the temperature at which the micrographs were taken were recorded.

##### Fourier-Transform Infrared Spectroscopy (FTIR)

The FTIR spectra of pure CBZ, MAL, SORB, XYL, PMs, and SD formulations were recorded using a Cary 630 FTIR Spectrometer (Agilent Technologies, Santa Clara, CA, USA). The FTIR spectra were obtained in absorbance mode between 650 and 4000 cm^−1^ with 16 scans and 4 cm^−1^ resolution.

##### Powder X-Ray Diffraction (PXRD)

The crystallinity or amorphous habit of the pure compounds, PMs, and SD formulations was investigated by qualitative PXRD analysis. The PXRD patterns were recorded using a Bruker D2 Phaser X-ray diffractometer (Bruker, Billerica, MA, USA) using Cu rays (λ = 1.54184 Å) at 30 kV and 30 mA current. The samples were packed onto a zero-background sample holder and analysed over a range of 4–40 °2Ɵ with a step width of 0.0162 °/s and a scan speed of 1 °/min.

#### 2.2.4. Determination of Drug Content

An amount equivalent to 100 mg CBZ was weighed from each resultant SD, dispersed in 50 mL methanol, and sonicated for 10 min to achieve a clear solution [2]. Each solution was diluted to 100 mL with distilled water to prepare a stock solution. One millilitre (1 mL) from the stock solution was further diluted to 100 mL with distilled water. A similar procedure was employed to prepare a CBZ reference standard solution with a concentration of 100 μg/mL.

Subsequently, the samples were analysed using high-performance liquid chromatography (HPLC). A Shimadzu (Kyoto, Japan) HPLC system consisting of a pump (model 515), an auto-sampler (ULTRA WISP 715), a UV detector (model 486), and the Millennium 32 software was utilised. CBZ was analysed at 285 nm using a Phenomenex Luna^®^ C18 reversed-phase column, 150 × 4.6 mm, 5 μm particle size (Torrance, CA, USA) at ambient temperature. A mobile phase consisting of methanol and ultrapure water in the ratio of 70:30 (*v*/*v*), at a flow rate of 1.2 mL/min and an injection volume of 20 μL injection volume was used [2]. Drug concentration was calculated by using the calibration curve in the range of 1–25 μg/mL, with a correlation coefficient (*r*^2^) of 0.998. The CBZ concentration was measured in triplicate and the mean ± SD were reported.

#### 2.2.5. Solubility Studies

Equilibrium solubility studies were carried out on CBZ, PMs, and the prepared SDs in distilled water maintained at 37 ± 0.5 °C. An excess amount of the CBZ and SDs were added to 5 mL distilled water in 10 mL glass polytop vials. The vials were sealed with Parafilm^®^ M (Bemis Inc., Neenah, WI, USA), sonicated for one hour, and subjected to agitation at 100 rpm using a magnetic stirrer bar for 72 h. Samples were collected at 24 h intervals, filtered through a 0.45 µm PVDF syringe filter into HPLC vials, and subsequently analysed using the described HPLC method. The solubility experiments were conducted in triplicate and the mean ± SD were reported.

#### 2.2.6. In Vitro Drug Release Studies

In vitro drug release studies were conducted to determine the performance of the PMs and SDs of CBZ in combination with the polyols (MAL, SOR, and XYL). The prepared PMs and SDs, equivalent to 100 mg of CBZ, were loaded into size 0 hard gelatine capsules and subjected to dissolution testing according to the USP dissolution test conditions for CBZ tablets [3]. The dissolution test was carried out using an Erweka DT 128 Light, United States Pharmacopeia (USP) type-I basket apparatus (Erweka GmbH, Langen, Germany), operated at 75 rpm, with 900 mL of a dissolution medium consisting of water containing 1% *w*/*v* sodium lauryl sulphate at 37 ± 0.5 °C. Aliquots of the dissolution medium (5 mL) were withdrawn at 5, 10, 15, 20, 30, 45, and 60 min. Fresh dissolution medium (5 mL) maintained at the same temperature was replaced after each withdrawal. The collected samples were further diluted (1:10) with the dissolution medium and analysed using the described HPLC method. The obtained data were analysed by calculating the amount of drugs released and the cumulative percentage of drugs released at the different time intervals.

#### 2.2.7. Dissolution Parameters

Data obtained from in vitro drug release studies were tested with the different model-independent techniques: mean dissolution time (MDT), dissolution efficiency (%DE), and similarity factor (f_2_) as detailed elsewhere [4,5,6]. MDT and %DE were used to compare the effect of different drug to polyol ratios as well as different types of polyols within the physical mixtures and solid dispersions. Similarity factor (f_2_) was calculated to compare the dissolution profiles of plain CBZ with the prepared physical mixtures and solid dispersions. An f_2_ value between 50 and 100 indicates similarity between the two profiles. The closer the f_2_ value to 100, the more similar or identical the release profiles and a decrease in the f_2_ value signifies a greater dissimilarity. All these parameters were calculated using the DD Solver software (add-in program for Microsoft Excel).

#### 2.2.8. Statistical Analysis

Statistical comparisons of the mean values of data were performed using one-way analysis of variance (ANOVA) with Microsoft Excel^®^ (Microsoft Corporation, Redmond, WA, USA) to calculate the *p*-values. The difference was deemed significant when the *p*-value was less than 0.05.

## 3. Results and Discussion

This study explored the possibility of preparing CBZ SDs using MAL, SOR, and XYL as co-formers or carriers. The solid-state and crystalline phase transformation of PMs and SDs of CBZ-polyol combinations were investigated using the acquired DSC, HSM, FTIR, and PXRD data.

### 3.1. Characterisation of CBZ-MAL SDs

The thermal behaviour and crystalline phase transformation of CBZ, CBZ control (heat–cool–heat sample), CBZ-MAL PMs, and SDs prepared with varying molar ratios of CBZ–MAL (2:1, 1:1, and 1:2) were investigated by DSC and the obtained thermograms are shown in Figure 2. The thermogram of pure CBZ showed a sharp first endothermic peak at 175.6 °C, corresponding with the melting of CBZ Form III, immediately followed by an exothermic peak at 178.5 °C, indicative of crystallisation to Form I and subsequently the melting of Form I at 192.8 °C [7]. These results are in agreement with the literature [7], confirming that the CBZ used in this study is the commercially available Form III. Similar results were observed for the CBZ control (not shown in Figure 2) with no significant changes in thermal events, corresponding with the published literature [8]. As depicted in Figure 2a, the melting point of the polyol, MAL, exhibited a single sharp endothermic peak at 153.6 °C, corresponding well with that reported in the literature [9].

The DSC thermograms observed for the PMs (Figure 2a) showed sharp endothermic peaks for both CBZ and MAL, indicating their crystalline nature. While no or an insignificant change was observed for MAL, the melting point of CBZ was reduced by ~10 °C for all three thermal events reported for Form III, with Form I melting at 181.8 °C (onset and end set at 179.9 °C and 185.8 °C, respectively). Similar results were reported previously for CBZ-XYL PMs [2], where the phenomenon was attributed to the presence of the polyol in the molten state, causing melting point depression of the higher melting point CBZ.

The DSC analyses of the prepared CBZ–MAL SDs (Figure 2b) showed a possible glass transition (*T_g_*) or *T_g_* with enthalpy relaxation visible at ~50 °C. This thermal event can be attributed to the *T_g_* of MAL as it was reported to be having a *T_g_* around 49 °C [10]. It is observed that the intensity of the relaxation peak increased with an increase in the MAL concentration and the absence of the melting peak of MAL indicated that the MAL remained amorphous in the SDs. A broad bell-shaped exothermic peak with an onset at ~103.2 °C and end set at ~115.2 °C signifies the crystallisation of CBZ fine crystallites, followed by a sharp melting peak at 180.0 °C, corresponding to the melting point of CBZ Form I, as observed in the case of the PMs. The presence of CBZ as fine crystallites can be confirmed by the change in enthalpy (Δ*H*), where the enthalpy needed for crystallisation is lower than the melting temperature (approximately half of the melting point enthalpy). This can be attributed to the increase in the Helmholtz free energy of the surface due to the decrease in the dimensions of the crystallites [11].

The results obtained for the SDs are in good correlation with the published investigations using the same preparation method [12,13]. This can be further confirmed by calculating the Δ*H*-values of each phase transition. It was reported that the Δ*H*-value associated with the first two events (melting of Form III, recrystallisation to Form I) accounts for approximately 12 J/g and the third event (melting of Form I) accounts for 100 J/g [12]. The experimental values obtained are recorded as 15 J/g and 91 J/g, which closely corresponds with those reported previously. These findings indicate that during the fusion method, CBZ may have recrystallised Form III into a possible amorphous MAL matrix followed by the recrystallisation of Form III to Form I during heating. The thermal analysis indicated that the SDs formed could potentially exist as two-phase glass suspensions with CBZ dispersed as fine crystalline particles.

In order to determine whether the prepared CBZ-MAL SDs exist as two-phase glass suspensions and whether any amorphization of MAL and/or CBZ occurred during the fusion method, further characterisation through HSM, FTIR, and PXRD was pursued. Figure 3 depicts the HSM micrographs obtained for CBZ-MAL(1:1)PM and CBZ-MAL SDs upon melting. The observations performed for CBZ-MAL(1:1)PM (Figure 3a) show the melting of MAL at 154 °C, corresponding to its melting point as stated above. The micrographs show unmolten material, which exhibited an onset of melting at 175 °C, immediately followed by small needle-like crystals growing outwards confirming the melting of CBZ Form III and the exothermic phase transition to Form I. These Form I crystals showed the onset of melting at 181 °C and complete melting was observed at 194 °C. The observed melting range is slightly higher in comparison with the DSC results. However, this can be expected as the DSC experiments were conducted under a more controlled environment in comparison with the HSM experimental conditions.

From the DSC and HSM results, it can be deduced that poor miscibility between MAL and CBZ exists, which could have a detrimental effect on the physical stability of potential solid dispersions (SDs). It is to be noted that the same phenomenon was observed for the other CBZ-MAL PMs (2:1 and 1:2), indicating the poor miscibility of CBZ in MAL. The observed poor miscibility was also confirmed by the DSC results, as depicted in Figure 2, which shows a clear distinction between the thermal events associated with MAL and CBZ. Similar results were presented by Djuris et al. [13] when the CBZ concentration was 50% w/w and above in a mixture of CBZ:Soluplus^®^.

The micrographs of CBZ-MAL SDs (Figure 3b–d) showed opaque appearances immediately after preparation, which further indicates the formation of solid crystalline suspensions. It was visually observed that crystallisation occurred throughout the whole sample, indicating the possible formation of a two-phase continuous SD. The HSM results were found to be in good correlation with the DSC results as the transition of the MAL from the glassy to the rubbery state due to enthalpy relaxation resulted in the formation of polygonal structures. The formation of these polygonal structures during heating is associated with the release of heat of crystallisation, characterised by an exothermic process, leading to the formation of solidified crystals. This was followed by a definitive recrystallisation process in which the formation of small nuclei was observed from ~105 °C, followed by the growth of fine hairlike crystals (marked with pointed arrows in Figure 3b–d, indicating the crystallisation of CBZ Form I. The morphology of the CBZ Form I observed during the HSM experiments corresponds with the reported literature [14]. The thermal behaviour observed for the CBZ–MAL SDs compared well with the DSC data (Figure 2) for all the thermal events, including the complete melting of CBZ Form I crystals at ~181 °C.

As reported in Figure 4, the PXRD spectra of the pure CBZ (Figure 4a) present with its characteristic peaks between 10 and 40 °2θ (peaks at 2θ = 13.22, 15.47, 19.72, 25.06, and 27.50°). These identified diffraction peaks correspond to the reported work for CBZ polymorph III [15,16]. The diffractogram obtained with MAL (Figure 4a) showed a characteristic high-intensity diffraction peak at 2θ = 23.42 °, signifying the crystalline nature.

The PXRD pattern for the prepared SDs (Figure 4b) resembled a typical amorphous halo. None of the characteristic peaks of CBZ Form III could be clearly identified. The fact that the prepared SDs showed either completely diffused or significantly diffused PXRD diffractograms suggests that the recrystallisation of CBZ Form III resulted in a significant reduction in the particle size of CBZ.

Rustichelli et al. [17] reported the FTIR bands characteristic of the known CBZ polymorphs. Form I is identified with the presence of bands at 3484, 1684, 1592, and 1393 cm^−1^; Form II with absorbance bands at 3473, 1673, 1591, and 1395 cm^−1^; and 3464, 1676, 1593, 1383, and 762 cm^−1^ for Form III. Form IV shows characteristic peaks at 3474, 1674, and 1394 with a small shoulder at 1418 cm^−1^ [15]. The FTIR spectrum of the CBZ obtained in this study (Figure 5) showed characteristic peaks at 3464 cm^−1^ (N–H stretching of the primary amine), 1674 cm^−1^ (C = O stretching), split peak at 1605 and 1593 cm^−1^ (C = C stretching and N–H deformation), 1378 cm^−1^ (C–H bending), and 760 cm^−1^ as the specific features of Form III [2,15,17,18]. The characteristic absorption peaks of MAL in the range of 3380–3257 cm^−1^ signify O-H stretching and C-H stretching functional groups. The FTIR spectra of the CBZ-MAL PMs (2:1 and 1:1) (Figure 5A) showed all the identified characteristic peaks of Form III, indicating no interaction between CBZ and MAL during the mere mixing of the two compounds. However, the CBZ-MAL(1:2)PM broadening in the wavenumber range of 3380–3257 cm^−1^ and reduced intensity of peaks 3464 cm^−1^ and 760 cm^−1^ indicate possible interaction with an increase in MAL concentration.

The FTIR spectra of all the CBZ-MAL SDs (Figure 5B) showed peak broadening in the wavenumber range of 3380–3257 cm^−1^ and the lack of characteristic absorbance bands signifying O-H stretching confirms the findings of DSC, indicating the amorphization of CBZ-MAL. The FTIR spectra of all the SDs showed significant peak broadening for the N–H stretching bands at 3464 cm^−1^, and a slight peak broadening at 1674 cm^−1^ (C = O stretching), indicating intermolecular interaction between CBZ and MAL. It is observed that the MAL concentration in the SDs had a direct impact on the peak broadening of the N-H stretching band, indicating a direct relation between the concentration of MAL and the intermolecular interactions with CBZ. The split peak at 1605 and 1593 cm^−1^ showed reduced intensity and appeared slightly merged. This may imply that the primary amine of CBZ either formed a weaker hydrogen bonding with the MAL [18] or the change could be due to reduced particle size. Interestingly, all the characteristic absorbance bands associated with CBZ Form III were identified in the FTIR spectra of all the SDs, and most importantly the absorbance peak at 760 cm^−1^ specifically attributed to this CBZ polymorphic form. No sharp well-defined characteristic absorbance bands for MAL were observed in the prepared SDs. These findings, thus, correlate with the DSC results and therefore, prove that CBZ exists as Form III dispersed in an amorphous MAL carrier system.

To confirm the preliminary conclusion that CBZ crystallised as Form III into an amorphous matrix of MAL immediately after preparation via fusion followed by the recrystallisation of CBZ Form I during the DSC and HSM analysis, a fresh sample of CBZ-MAL(1:1)SD was prepared with the fusion method on the HSM. The prepared SD was heated at 10 °C/min until the crystallisation point was observed (105 °C). Subsequently, the temperature of the heating stage was maintained isothermally for an hour to allow recrystallisation under isothermal conditions. The resultant crystals were subjected to the PXRD analysis to determine the formed CBZ polymorph. As depicted in Figure 6, the characteristic peaks of Form I were observed at °2θ = 6.1, 7.9, 9.3, 12.2, 13.1, and 19.8 [7,17]. These results confirmed the observed thermal behaviour. Since CBZ Form III was identified in all the CBZ-MAL SDs via FTIR analyses as well as the confirmation of the recrystallisation of Form I during the heating of all the CBZ-MAL SDs, it was concluded that during the fusion method, CBZ Form III gets entrapped into an amorphous matrix of MAL.

### 3.2. Characterisation of CBZ-SOR SDs

The thermogram of D-sorbitol (SOR) (Figure 7a), presented with a small endothermic peak at 74.8 °C and a sharp endothermic peak at 98.8 °C, with an enthalpy of fusion (Δ*H*) of 162.5 J/g corresponding to the melting point of polymorphic Form B of SOR [16,19]. Jeganathan and Prakya [16], attributed the endothermic peak at 75 °C to the presence of D-sorbitol in the amorphous form. The DSC scans of CBZ-SOR PMs (Figure 7a) showed the presence of both prominent peaks of the polyol with no significant change in melting point. However, a significant reduction in the melting point of CBZ (~20 °C) for all three CBZ-associated events was reported for Form III, with Form I melting at ~170.0 °C.

In the case of the SDs (2:1, 1:1, and 1:2) (Figure 7b), a small endothermic event was observed at 45 °C (as indicated by red box), and the lack of the characteristic melting peak of SOR indicates the complete amorphization of the polyol. However, the relaxation of SOR at a lower temperature could be a potential indication of the physical instability of the SDs. Similarly to the observations made for CBZ-MAL SDs, a broad exothermic peak at 95 °C (onset at ~89 °C and end set ~99 °C), followed by a sharp melting peak at ~168 °C, corresponding to the melting point of CBZ Form I. The significant reduction in the melting point is an indication of the reduction in the crystal lattice energy [20], which has been studied for its positive impact on the solubility and dissolution of drugs [21]. The thermal analysis indicates that the SDs formed with SOR are similar in solid-state nature as exhibited by the CBZ-MAL SDs, where a two-phase glass suspension is formed with CBZ dispersed as fine crystalline particles in amorphous SOR.

The HSM micrographs obtained for CBZ-SOR(1:1)PM (Figure 8a) correlate with the DSC results, where SOR and CBZ melt at their respective melting points, with CBZ melting and recrystallising (Form I) at 178 °C (pointed arrow for needle-like crystals developed), followed by complete melting at 194 °C. This indicates poor miscibility between CBZ and SOR similar to the CBZ-MAL mixtures (SDs).

The micrographs of CBZ-SOR SDs (Figure 8b–d) showed highly opaque appearances, indicating the formation of continuous solid crystalline suspensions. The HSM results are in good correlation with the DSC, where the first endothermic event reported on the DSC thermograms at 45 °C was observed at ~60 °C. This event can be attributed to the melting of amorphous SOR as observed visually, where the molten material appeared less viscous and flowing (molten liquid is highlighted in Figure 8c,d at 63 and 60 °C, respectively). This argument can further be substantiated by the fact that SOR exhibits a *T_g_* below zero degrees (−5.15 °C) [22]. The resultant of SOR melting resulted in the increased proximity of the CBZ crystals, resulting in crystal growth. During the HSM analyses, all three SDs showed a visual thickening of the sample at ~90 °C, signifying nucleation and crystal growth. This can be attributed to the fact that the CBZ is dispersed as fine crystalline particles in amorphous SOR, and the thermal behaviour of polymorphic drugs with a change in temperature could result in a high crystallisation tendency [23,24]. The thermal behaviour observed for the CBZ–SOR SDs compared well with the DSC data (Figure 7) for all the thermal events, including the complete melting of CBZ Form I crystals at ~175 °C.

The PXRD spectra of the SOR (Figure 9) present with its characteristic peak at 2θ = 19.13 °2θ) [25] and several high-intensity diffraction peaks, signifying the crystalline nature. The PXRD pattern for the SD prepared with different molar ratios of SOR resembled a typical amorphous halo with several low-intensity peaks, highlighting a significant reduction in the particle size of both CBZ and SOR.

The FTIR spectra of SOR (Figure 10) show its characteristic absorption peaks in the 3000 and 3500 cm^−1^ region, signifying the O-H stretching, C-H stretching at 2930 cm^−1^, O-H bending at 1298 cm^−1^, and C-O stretching vibrations between 950 and 1150 cm^−1^ [26,27], while the FTIR spectrum of the CBZ-SOR(1:1)PM (Figure 10A) showed all the identified characteristic peaks of Form III, indicating no interaction between CBZ and SOR. A similar trend is seen with the FTIR spectra of all the CBZ-SOR SDs (Figure 10B), where all the characteristic peaks of CBZ Form III could be identified. However, these peaks were found to be significantly broadened or reduced in intensity, indicating weak intermolecular interactions between CBZ and SOR.

### 3.3. Characterisation of CBZ-XYL SDs

The DSC curves presented in Figure 11a present a sharp endothermic peak for XYL at 98 °C, which corresponds with the literature reports [2,28]. Interestingly, the PMs of CBZ-XYL show peak broadening for XYL. The CBZ-XYL PMs also presented with less intense, significantly broadened peaks that are shifted to lower melting points. From Figure 11a, it can be seen that the first and second thermal events (melting of CBZ Form III and recrystallisation to Form I) disappeared with an increase in XYL concentration and even for the CBZ-XYL(2:1) PM, these three thermal events were almost indistinguishable. Similar results were reported for CBZ-XYL PMs and SDs prepared by a hot melt extrusion process [2]. Such a phenomenon can be attributed to the miscibility of the drug in the carrier [29]. However, the lack of complete disappearance of the melting peaks among the CBZ-XYL PMs can be attributed to the partial miscibility of the drug in the carrier. However, a significant reduction in the melting point of CBZ (~30 °C) for all three events reported for Form III, with Form I melting at ~163.0 °C was observed.

Similarly to the CBZ SDs of MAL and SOR, the CBZ-XYL SDs also exhibited recrystallisation at ~85 °C, represented by an exothermic peak visible on the thermograms (Figure 12b). The results obtained for the CBZ-XYL SDs followed a similar trend seen among the PMs, where large endothermic peaks at significantly lower temperatures were observed. The CBZ-XYL(2:1)SD showed a broad endothermic peak with a dissimilar peak shape in comparison with the two higher XYL ratio SDs. This could be considered as an indication of a broad size distribution of the crystallites in the SD. The broadness reduced with an increase in the XYL concentration, represented by relatively sharp endothermic peaks. The significant reduction in the melting point (from 192.8 to ~160.0 °C) is a potential indication of an increase in the solubility and dissolution rates. The thermal analysis indicated that the SDs formed with XYL may be characterised as two-phase continuous glass suspensions.

The HSM micrographs (Figure 12) correlated very well with the DSC results. It can be seen that the CBZ-XYL(1:1)PM (Figure 12a) showed different melting behaviour to the other polyols as the melting of CBZ takes place at a significantly lower temperature (160 °C), indicating good miscibility between CBZ and XYL.

The opaque appearance of the CBZ-XYL SDs, as observed by means of microscopy, further confirms the solid-state nature of the SDs. Interestingly, a melting event was observed (pointed arrows in red colour), which was not observed through the DSC analyses and shows a resemblance to the melting of SOR as seen previously with the CBZ-SOR SDs. Subsequently, an increase in CBZ crystal growth (pointed arrows in yellow colour in Figure 12b–d) was observed, which was visually observed through the increase in the turbidity of the sample. The HSM micrographs presented for ~90 °C show the thickening of the sample (increase in opaqueness), signifying the crystal growth.

The PXRD spectra of XYL (Figure 13) present with its characteristic high-intensity diffraction peaks at °2θ = 24.294, 31.85, and 38.14 [30], signifying the crystalline nature. Unlike the PXRD patterns observed for the SDs of MAL and SOR (two halo humps observed), a single halo hump was observed for all the CBZ-XYL SDs. This observation can further support the argument of good miscibility between CBZ and XYL. 

The FTIR spectrum of XYL (Figure 14) corresponds with the published literature, showing its characteristic broad peaks at 3100–3400 cm^−1^ for the –OH stretch, 1165 cm^−1^ for –C = O stretch, and 1050–1300 cm^−1^ for the carboxylic acid and alcohol groups moieties [2,31]. While the FTIR spectra for CBZ-XYL PMs retained all the characteristic peaks without any broadening or reduction intensity (Figure 14A), the FTIR spectra of all the CBZ-XYL SDs (Figure 14B) showed the characteristic peaks of CBZ Form III and XYL at a reduced peak intensity, indicating weak intermolecular interaction between CBZ and XYL.

### 3.4. Equilibrium Solubility Studies

The CBZ content in the PMs and SDs was found to be between 94.3 ± 2.4 to 102.7 ± 1.8%, which meets the prescribed limits of 92 to 108% of the USP monograph for CBZ tablets [3]. The equilibrium solubility of CBZ, CBZ–polyol PMs, and CBZ–polyol SDs was investigated and compared with the equilibrium aqueous solubility of pure CBZ in distilled water at 24, 48, and 72 h. The aqueous solubility of the raw material CBZ was 231.8 ± 4.8 µg/mL and 233.7 ± 4.6 µg /mL at 48 and 72 h, respectively (Table 1). The results obtained for CBZ are in good correlation with the published data [32,33].

The solubility of CBZ slightly increased among all the PMs with the highest increase of 18% for CBZ-XYL(1:2)PM and the lowest of 0.5% for CBZ-MAL(2:1)PM. Amongst the PMs, an increase in solubility was associated with increasing the concentration of the polyol. The solubility of CBZ among the SDs increased substantially in the order of CBZ-XYL SDs (6.4-fold) > CBZ—SOR (2.9-fold) > SDS CBZ—MAL SDS (2-fold). Except for CBZ-MAL(1:2)SD, the solubility of the CBZ among all the SDs increased with an increase in the concentration of the polyol. The anomalous behaviour of the CBZ-MAL(1:2)SD can be explained based on the extremely high solubility of MAL in water (175% *w*/*w*) [34]; resulting in MAL solubilising quicker, saturating the fixed volume used for solubility testing, and thus, detrimentally affecting the CBZ solubility [35,36]. Further to this, an increase in the viscosity of the solutions with an increase in MAL concentration was observed, which could have also affected the solubility of CBZ [37].

The solubility enhancement among the PMs and SDs can be attributed to the enhanced wettability of the drug particles facilitated by the hydrophilic sugar carriers and the reduced particle size [38,39]. The partial miscibility observed between CBZ and XYL from the DSC studies could explain the significant increase in solubility for the CBZ-XYL SDs (*p* < 0.01) over SDs prepared with MAL and SOR. Similar results were reported in the literature, where predictions made using in silico molecular dynamic simulation correlated with the experimental results of drug carrier miscibility effect on improving the amorphous nature of the SDs and improved solubility [40].

The physicochemical properties, such as the number of hydroxyl (-OH) groups along its carbon chain, play a central role in solubility by forming hydrogen bonds with drug molecules. However, the structure of these polyols can enhance or reduce solubility depending on factors like molecular size and molecular weight, where larger molecules may face steric hindrance and experience less efficient interaction with solvent molecules. In addition, polyols that are linear or acyclic tend to have greater solubility in water over the cyclic polyols [41,42]. This can further explain the high solubility of CBZ in the presence of xylitol and sorbitol over maltitol. Despite maltitol possessing a large number of hydroxyl groups (9), the cyclic nature of the compound and its high molecular weight (344.31 g/mol) seem to have a negative effect on solubility.

Observations made between sampling points indicated that 93 to 99% of the drug is solubilised within 24 h, followed by 99 to 100% at 48 h. It is also observed that there were instances of reduced drug content at 72 h with the highest loss of 6.2 µg/mL in the case of CBZ-XYL(1:2)SD. However, the change was found to be insignificant (*p* = 0.12), indicating the lack of solution-mediated phase transition among all the SDs studied.

### 3.5. In Vitro Drug Release

The CBZ release profiles from SDs prepared with MAL, SOR, and XYL are shown in Figure 15. From the graphs presented, the CBZ raw material showed incomplete dissolution with 52% within 60 min. The percentage drug release observed at 60 min for PMs of CBZ-MAL (Figure 15a), CBZ-SOR (Figure 15b), and CBZ-XYL (Figure 15c) indicated a slight increase in comparison with pure CBZ. The CBZ-XYL PMs showed slightly higher drug release than PMs of MAL and SOR. However, the differences seen among the PMs of various polyol combinations were found to be statistically insignificant (*p* = 0.68). In contrast to the PMs, all the SDs reported increased drug release (%), where all the SDs released more than 75% within 60 min. The drug release for SDs is in the order of CBZ-XYL (99.8% ± 0.44) > CBZ-SOR (87.8% ± 2.2) > CBZ-MAL (77.5% ± 1.8). These results correlate well with the solubility results reported for SDs. From the graph, it can be observed that the drug release is high for all the SDs across all the sampling points.

The results of the calculated %DE_60 min_ (Table 2) exhibited the minimum for pure CBZ (42%). The %DE for all the PMs was found to be close to the pure drug with higher values reported in the case of CBZ-XYL combinations (48–52%). However, the ANOVA results revealed that there was no statistically significant difference (*p* = 0.06) with a change in the polyol concentration as well as the type of the polyol. Similarly to the PMs, there was no statistically significant difference (*p* = 0.1) with a change in the polyol concentration among SDs. However, the type of polyol used had a significant effect (*p* < 0.01) and the % DE was found to be in the order for XYL > SOR > MAL with the maximum %DE showed for CBZ-XYL(1:1)SD (91%). The results indicate that SDs are more effective in increasing the dissolution rate than PMs.

The MDT for PMs were found to be the longest, similar to pure CBZ (11.3 min), ranging between 10.2 and 12.9 min with no statistically significant differences (*p* = 0.22). The SDs found to have the lowest MDT of 5.2 (CBZ-XYL(1:1)SD), 8.2 (CBZ-MAL(1:1)SD) and 8.8 (CBZ-SOR(1:2)SD), indicating higher drug releasing capacity of the SDs over the pure drug and PMs, with XYL-based SDs being the superior (*p* < 0.01). It is also interesting to see that there is no statistically significant difference (*p* = 0.97) in drug release with a change in polyol concentration for all the polyols studied.

The percent drug release was also compared by similarity factor (f_2_) using the pure CBZ as a reference (Table 2). Irrespective of the polyol used, the dissolution rates of all the PMs were found to be similar to the pure drug, where the f_2_ values were found to be more than 50 (52.7–73.1). A trend of decreasing f_2_ values (<50) for all the SDs was observed in the order of XYL < SOR < MAL. The dissolution rates of all the SDs significantly exceeded the plain drug as indicated by the f_2_ values (16.1–33.2). Similarly to the %DE and MDT values reported, the polyol concentration has no statistically significant effect on the percent drug release for PMs (*p* = 0.5) and SDs (*p* = 0.1).

The superior performance of XYL observed over other polyols is well reported [2,31]. This is mainly attributed to its high aqueous solubility, leading to the creation of a hydrophilic environment for improved solubilization of poorly soluble drugs, and hence, faster dissolution rates [43]. Among the polyols studied, sorbitol has higher aqueous solubility than xylitol and maltitol [44]. The anomaly observed with regard to the lesser dissolution rate of CBZ-SOR SDs over CBZ-XYL SDs can be explained by the increase in viscosities and densities of solutions with increased molecular weight due to the presence of the high number of carbon molecules (maltitol-12, sorbitol-6, and xylitol-5) [45,46]. The possible mechanisms attributed to the increased dissolution rate of polyol SDs are mainly the increased wettability of the drug by creating a hydrophilic environment around the drug and reduced interfacial tension [6,47]. In addition, the fusion process could have resulted in the dispersion of the drug at a molecular level, availing a high surface area for contact with the dissolution medium. The dissolution results obtained correlate well with the observations and confirm the hypothesis made during solid-state characterisation.

## 4. Conclusions

CBZ SDs were prepared using three crystalline hydrophilic carriers; MAL, SOR, and XYL by the fusion method. The prepared SDs were comprehensively characterised to determine the solid-sate nature of the SDs. The DSC and HSM results indicated that the SDs prepared were glass suspensions with the carriers existing in the amorphous state and the CBZ suspended as very fine crystallites throughout the amorphized polyol carrier. The fine crystalline nature of the CBZ is confirmed by the halo appearance of PXRD diffractograms and through the SEM image analysis. The FTIR studies showed that the CBZ is present in its thermodynamically stable polymorph, Form III, and formed weak intermolecular interactions with the polyols. The equilibrium solubility and in vitro dissolution studies showed enhanced aqueous solubility and increased dissolution rate for all the SDs with comparatively better performance exhibited by the CBZ-XYL SDs. The results obtained explain the possible extrapolation of preparing polyol-based SDs of BCS Class II drugs with good glass-forming ability for enhanced solubility and dissolution.

## Figures and Tables

**Figure 1 pharmaceutics-17-00321-f001:**
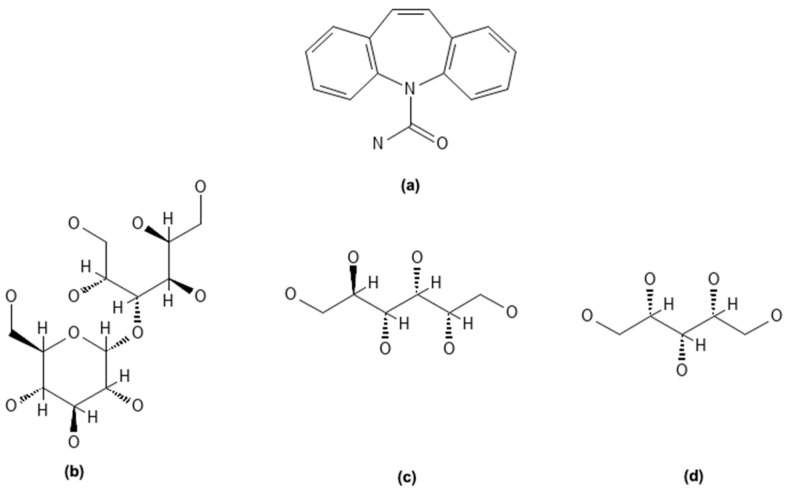
Molecular structures of (**a**) CBZ, (**b**) MAL, (**c**) SORB, and (**d**) XYL.

**Figure 2 pharmaceutics-17-00321-f002:**
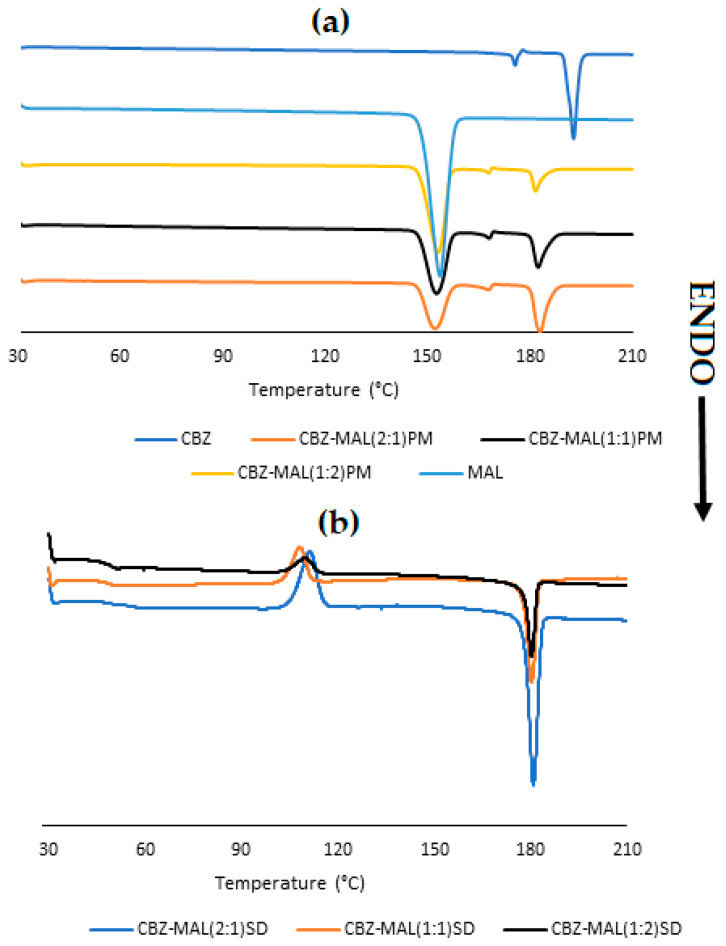
Overlay of DSC thermograms obtained for (**a**) MAL, CBZ, and CBZ-MAL PMs and (**b**) CBZ-MAL SDs.

**Figure 3 pharmaceutics-17-00321-f003:**
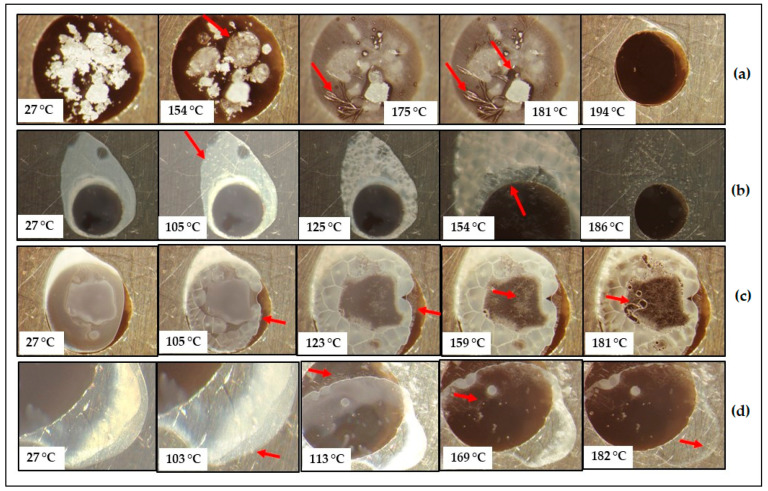
HSM micrographs obtained for the prepared (**a**) CBZ–MAL(1:1)PM, (**b**) CBZ–MAL(2:1)SD, (**c**) CBZ–MAL(1:1)SD, and (**d**) CBZ–MAL(1:2)SD during heating at 10 °C/min.

**Figure 4 pharmaceutics-17-00321-f004:**
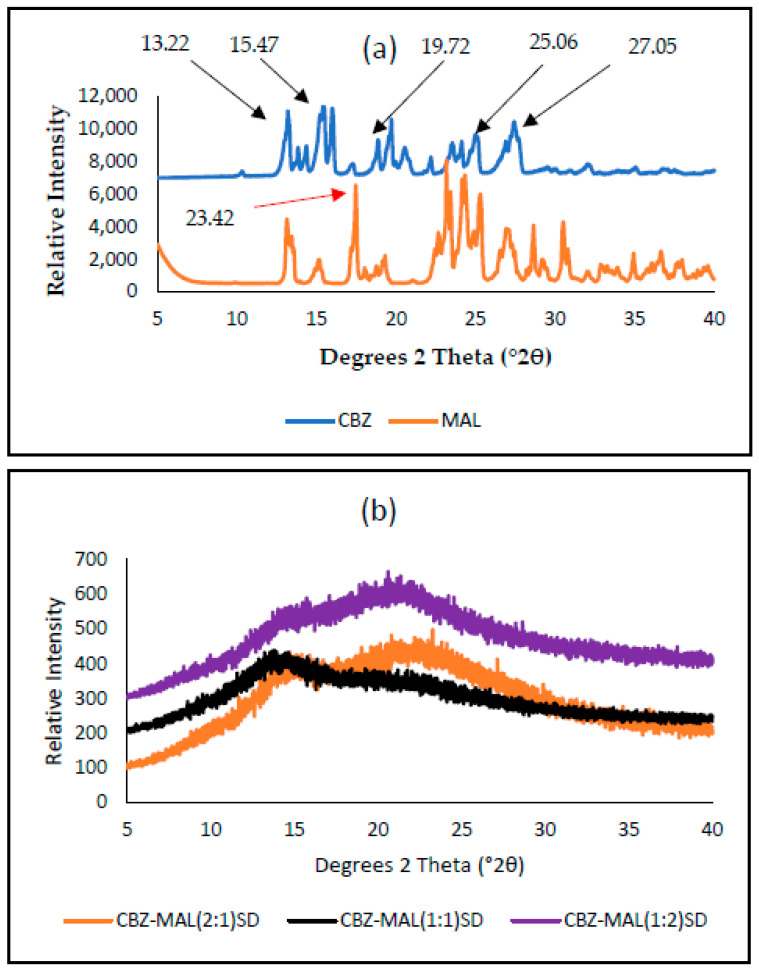
PXRD diffractograms obtained for pure CBZ, MAL (**a**) and the prepared SDs of CBZ–MAL in the molar ratios of 2:1, 1:1, and 1:2 (**b**).

**Figure 5 pharmaceutics-17-00321-f005:**
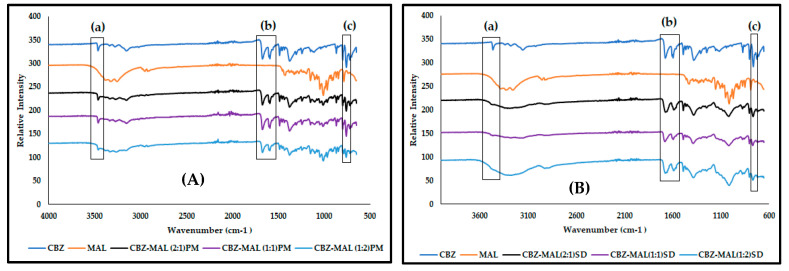
Overlay of the FTIR spectra obtained for (**A**) CBZ and MAL in comparison with the CBZ–MAL PMs and (**B**) all CBZ–MAL SDs, with (a) indicating the observed peak broadening in the 3464 cm^−1^ and (b) 1674 cm^−1^ wavenumber regions and (c) denoting the absorbance band at 760 cm^−1^, considered characteristic of CBZ Form III.

**Figure 6 pharmaceutics-17-00321-f006:**
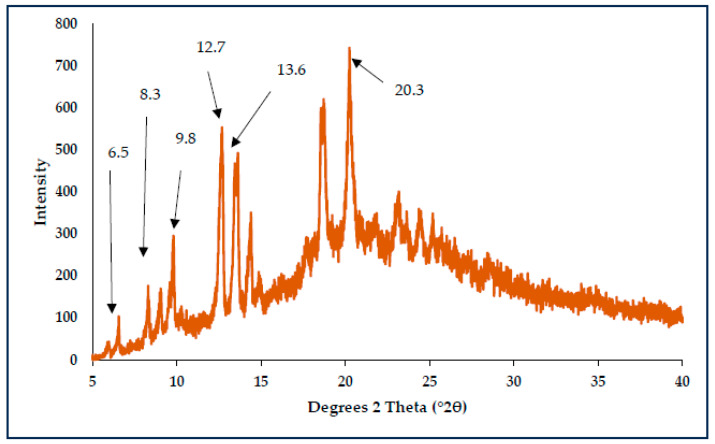
PXRD diffractogram for CBZ form I crystals formed under isothermal conditions of recrystallisation.

**Figure 7 pharmaceutics-17-00321-f007:**
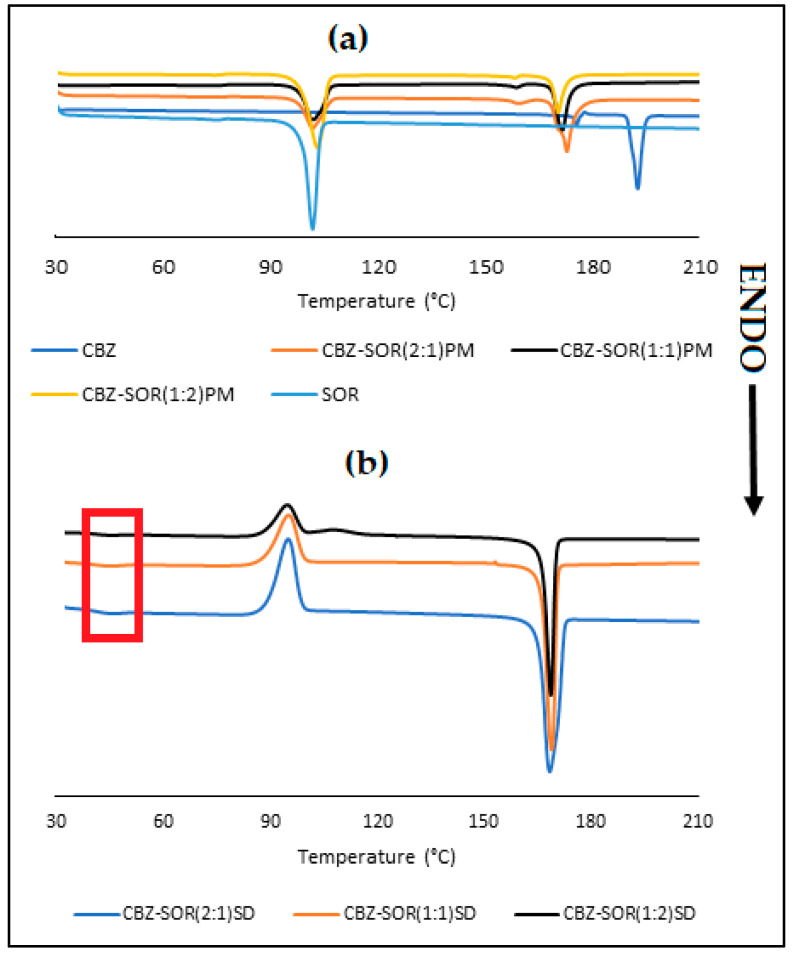
Overlay of DSC thermograms obtained for (**a**) SOR, CBZ, and CBZ-SOR PMs and (**b**) CBZ-SOR SDs.

**Figure 8 pharmaceutics-17-00321-f008:**
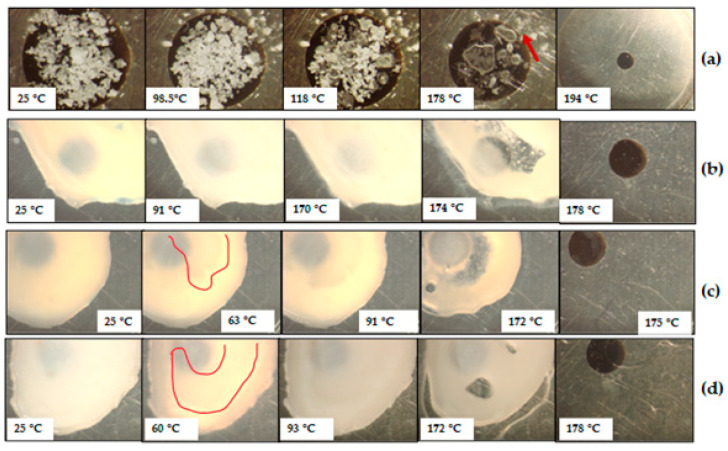
HSM micrographs obtained for the prepared CBZ–SOR(1:1)PM (**a**), CBZ–SOR(2:1)SD (**b**), CBZ–SOR(1:1)SD (**c**), and CBZ–SOR(1:2)SD (**d**) during heating at 10 °C/min.

**Figure 9 pharmaceutics-17-00321-f009:**
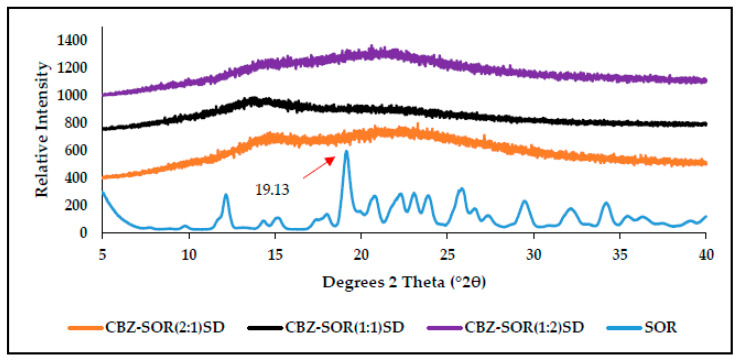
PXRD diffractograms obtained for pure SOR and the prepared SDs of CBZ–SOR in the molar ratios of 2:1, 1:1, and 1:2.

**Figure 10 pharmaceutics-17-00321-f010:**
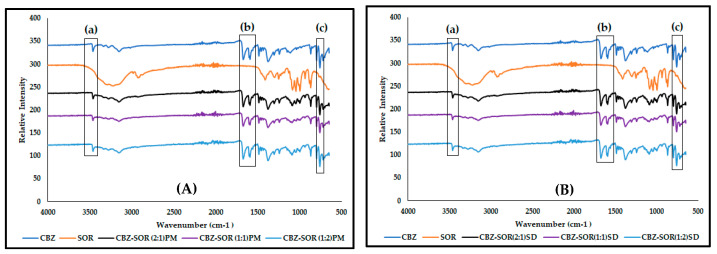
Overlay of the FTIR spectra obtained for CBZ and SOR in comparison with (**A**) the CBZ–SOR PMs and (**B**) CBZ–SOR SDs, with (a) indicating the observed peak broadening in the 3464 cm^−1^ and (b) 1674 cm^−1^ wavenumber regions and (c) denoting the absorbance band at 760 cm^−1^, considered characteristic of CBZ Form III.

**Figure 11 pharmaceutics-17-00321-f011:**
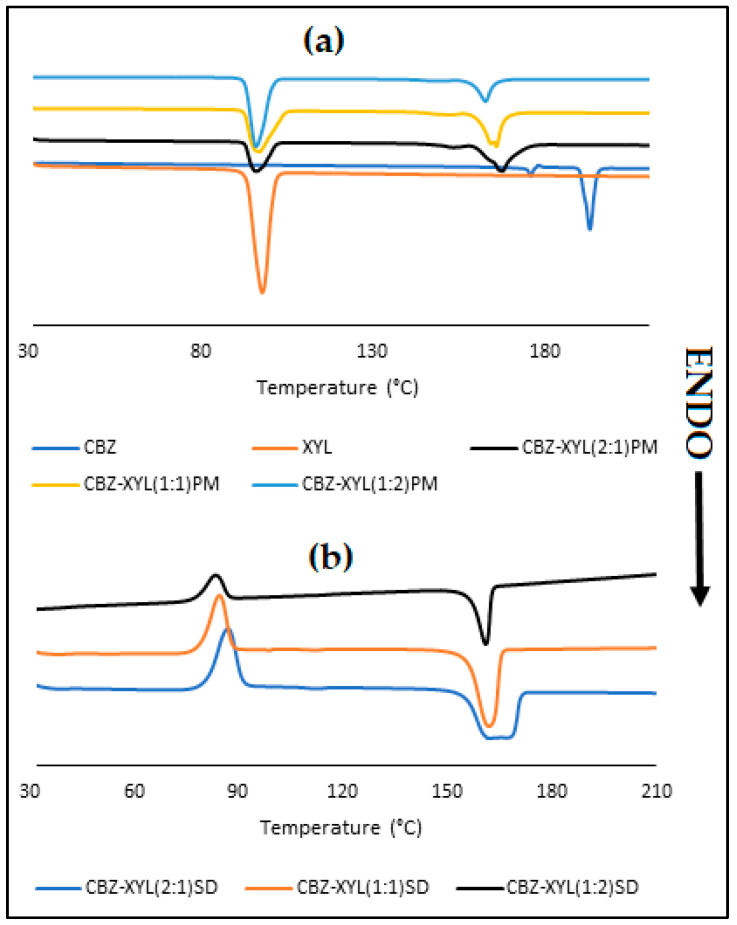
Overlay of DSC thermograms obtained for (**a**) XYL, CBZ, CBZ-XYL PMs and (**b**) CBZ-XYL SDs.

**Figure 12 pharmaceutics-17-00321-f012:**
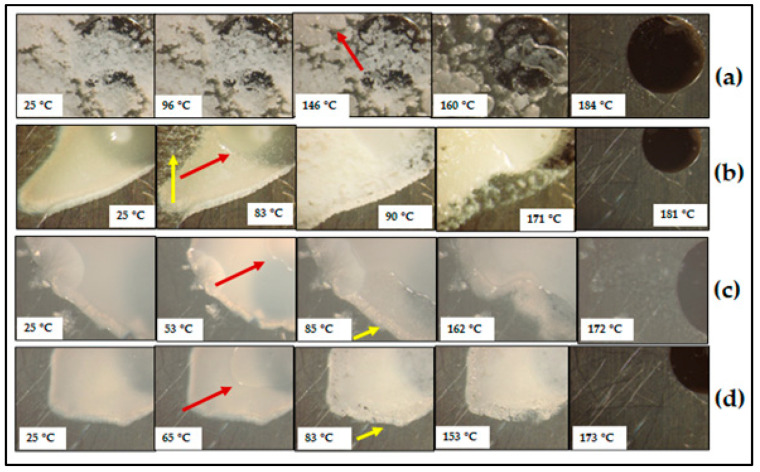
HSM micrographs obtained for the prepared (**a**) CBZ–XYL(1:1)PM, (**b**) CBZ–XYL(2:1)SD, (**c**) CBZ–XYL(1:1)SD, and (**d**) CBZ–XYL(1:2)SD during heating at 10 °C/min.

**Figure 13 pharmaceutics-17-00321-f013:**
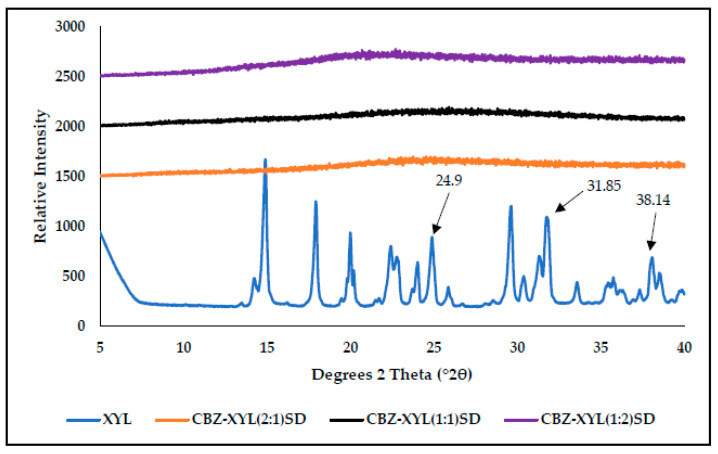
PXRD diffractograms obtained for pure XYL and the prepared SDs of CBZ–XYL in the molar ratios of 2:1, 1:1, and 1:2.

**Figure 14 pharmaceutics-17-00321-f014:**
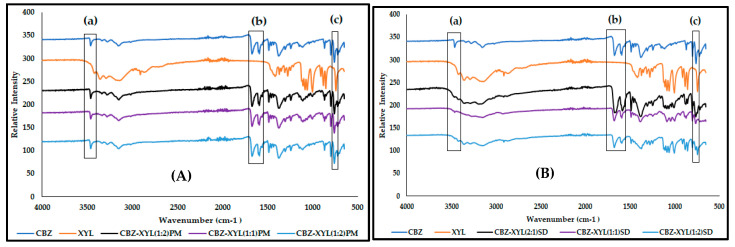
Overlay of the FTIR spectra obtained for CBZ and XYL in comparison with (**A**) the CBZ–XYL PMs and (**B**) CBZ–XYL SDs, with (a) indicating the observed peak broadening in the 3464 cm^−1^ and (b) 1674 cm^−1^ wavenumber regions and (c) denoting the absorbance band at 760 cm^−1^, considered characteristic of CBZ Form III.

**Figure 15 pharmaceutics-17-00321-f015:**
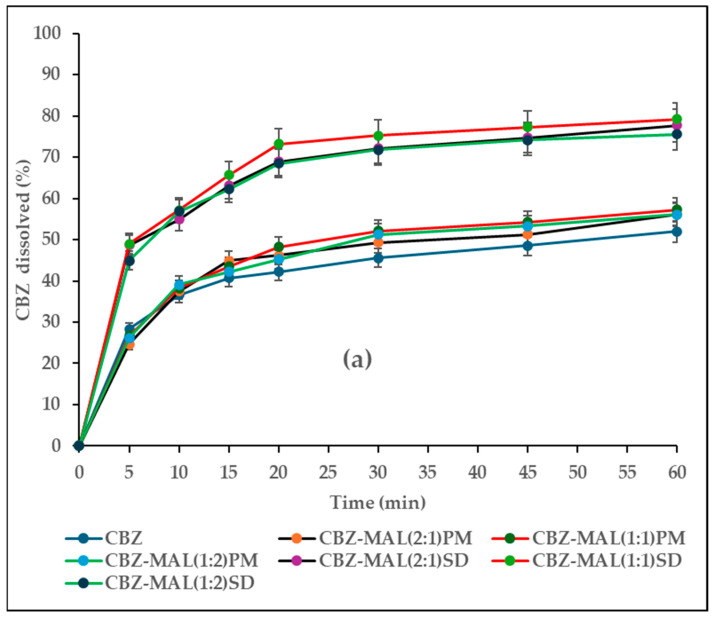
(**a**–**c**)In vitro release profiles of pure CBZ, PMs, and SDs in combination with polyols in varying molecular ratios.

**Table 1 pharmaceutics-17-00321-t001:** Summary of equilibrium solubility of CBZ, PMs, and SDs in combination with polyols in varying molecular ratios.

Sample ID	24 h (µg/mL)	48 h (µg/mL)	72 h (µg/mL)	Solubility Increase (%)
	Mean ± SD, n = 3			
CBZ	216.3 ± 5.3	231.8 ± 4.8	233.7 ± 4.6	-
CBZ-MAL(2:1)PM	228.9 ± 6.1	233.0 ± 5.4	231.6 ± 3.1	0.5
CBZ-MAL(1:1)PM	242.3 ± 4.9	243.3 ± 4.1	243.8 ± 3.5	5.0
CBZ-MAL(1:2)PM	255.2 ± 3.7	262.4 ± 3.2	259.3 ± 3.6	13.2
CBZ-MAL(2:1)SD	302.8 ± 3.3	312.7 ± 4.1	310.2 ± 3.2	34.9
CBZ-MAL(1:1)SD	475.9 ± 4.1	477.2 ± 5.7	476.9 ± 5.6	105.9
CBZ-MAL(1:2)SD	454.3 ± 3.1	460.3 ± 4.7	458.6 ± 3.4	98.6
CBZ-SOR(2:1)PM	242.6 ± 4.8	247.5 ± 4.0	248.1 ± 3.9	6.8
CBZ-SOR(1:1)PM	244.4 ± 5.6	248.1 ± 3.1	248.3 ± 4.6	7.0
CBZ-SOR(1:2)PM	261.7 ± 5.7	260.8 ± 5.8	260.6 ± 5.1	12.5
CBZ-SOR(2:1)SD	475.9 ± 4.1	477.2 ± 5.7	476.9 ± 5.6	105.9
CBZ-SOR(1:1)SD	605.6 ± 6.6	612.0 ± 6.8	612.3 ± 5.2	164.0
CBZ-SOR(1:2)SD	670.5 ± 4.8	670.7 ± 5.5	671.6 ± 5.5	189.3
CBZ-XYL(2:1)PM	246.2 ± 4.7	248.4 ± 3.9	248.2 ± 4.3	7.2
CBZ-XYL(1:1)PM	260.4 ± 5.9	263.0 ± 5.6	263.6 ± 4.2	13.5
CBZ-XYL(1:2)PM	268.1 ± 6.7	273.5 ± 5.1	272.1 ± 5.8	18.0
CBZ-XYL(2:1)SD	1341.2 ± 8.1	1426.1 ± 8.6	1428.6 ± 7.3	515.2
CBZ-XYL(1:1)SD	1414.6 ± 9.6	1436.8 ± 8.7	1433.4 ± 5.9	519.5
CBZ-XYL(1:2)SD	1481.8 ± 7.3	1488.2 ± 7.6	1482.5 ± 5.1	541.9

**Table 2 pharmaceutics-17-00321-t002:** Dissolution parameters of carbamazepine from PMs and SDs.

Sample ID	MDT (min)	%DE	f_2_
CBZ	11.3	42	-
CBZ-MAL(2:1)PM	12.2	45	73.1
CBZ-MAL(1:1)PM	11.1	47	67.4
CBZ-MAL(1:2)PM	11.2	46	72.3
CBZ-MAL(2:1)SD	9.1	66	32.4
CBZ-MAL(1:1)SD	8.2	68	30.3
CBZ-MAL(1:2)SD	8.3	65	33.2
CBZ-SOR(2:1)PM	12.9	45	74.0
CBZ-SOR(1:1)PM	12.6	46	71.1
CBZ-SOR(1:2)PM	12.1	49	60.1
CBZ-SOR(2:1)SD	9.7	71	27.7
CBZ-SOR(1:1)SD	10.5	73	27.0
CBZ-SOR(1:2)SD	8.8	76	23.8
CBZ-XYL(2:1)PM	12.2	48	63.4
CBZ-XYL(1:1)PM	10.9	50	56.9
CBZ-XYL(1:2)PM	10.2	52	52.7
CBZ-XYL(2:1)SD	5.5	90	16.6
CBZ-XYL(1:1)SD	5.2	91	16.1
CBZ-XYL(1:2)SD	5.9	90	16.7

## Data Availability

The data collected in this study are presented and available in this article.

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
