# Peer review of "An Investigation into the Effect of Maltitol, Sorbitol, and Xylitol on the Formation of Carbamazepine Solid Dispersions Through Thermal Processing"

_pharmaceutics, 2025, doi:10.3390/pharmaceutics17030321_

Round 1
Reviewer 1 Report
Comments and Suggestions for Authors
Review 1 Pharmaceutics-3440823
The study entitled “An investigation into the effect of maltitol, sorbitol and xylitol on the formation of carbamazepine solid dispersions through thermal processing” by Poka et al. is an interesting scientific work to improve the dissolution profiles of poorly soluble drugs. The authors evaluate two different polyols maltitol (MAL), sorbitol (SOR) and xylitol (XYL) as hydrophilic crystalline carrier para prepared carbamazepine solid dispersions. The XYL SDs showed the best solubility and dissolution rate results.
The work is clear and well written. We consider that this work is suitable for publication in the journal "Pharmaceutics". However, we have included some minor considerations that could improve the quality of this publication.
Minor changes
Metodos
2.2.3.3 Fourier-transform infrared spectroscopy (FTIR)
Line. 139-142
This paragraph is not necessary in this methods section
The spectra were analysed for change in intensity or absence or shift in the wave numbers of the characteristic peaks to determine any possible interactions between CBZ and the selected polyols in the PMs and SDs.
Results
1- Please review the entire text to write all formulations the same way (eg. without space)
Line 267. CBZ-MAL(1:1) PM it should be CBZ-MAL(1:1)PM. Check the whole text
3.4 Equilibrium Solubility Studies
2-Line 558-559
It is necessary to indicate the medium used
were investigated and compared with the equilibrium aqueous solubility of pure CBZ in distilled water at 24, 48 and 72 hours.
3-Line 575-579
You have to include references in these paragraphs
resulting in MAL solubilising quicker, saturating the fixed volume used for solubility testing and, thus, detrimentally affecting the CBZ solubility [ ref XX]. Further to this, an increase in viscosity of 577 the solutions with an increase in MAL concentration was observed, which could 578 have also affected the solubility of CBZ [ ref XX].
Author Response
Dear reviewer, thank you for the comments. Please see the point by point response.
Comment 1:
2.2.3.3 Fourier-transform infrared spectroscopy (FTIR)
Line. 139-142
This paragraph is not necessary in this methods section
The spectra were analysed for change in intensity or absence or shift in the wave numbers of the characteristic peaks to determine any possible interactions between CBZ and the selected polyols in the PMs and SDs.
Response: Thank you very much for the comment. As recommended, this part is deleted.
Comment 2:
1- Please review the entire text to write all formulations the same way (eg. without space)
Line 267. CBZ-MAL(1:1) PM it should be CBZ-MAL(1:1)PM. Check the whole text
Response: Thank you very much for the comment. This is checked across the document and changes made according to the recommendation to be consistent. Changes made now reflects in Lines 268 and 431.
Comment 3:
3.4 Equilibrium Solubility Studies
2-Line 558-559
It is necessary to indicate the medium used
were investigated and compared with the equilibrium aqueous solubility of pure CBZ in distilled water at 24, 48 and 72 hours.
Response: Thank you very much for the comment. The medium “distilled water” was added to the discussion (Line 573).
Comment 4:
3-Line 575-579
You have to include references in these paragraphs
resulting in MAL solubilising quicker, saturating the fixed volume used for solubility testing and, thus, detrimentally affecting the CBZ solubility [ ref XX]. Further to this, an increase in viscosity of 577 the solutions with an increase in MAL concentration was observed, which could 578 have also affected the solubility of CBZ [ ref XX].
Response: Thank you very much for the comment. References are now added to support the statements. (Lines 589-593).
“resulting in MAL solubilising quicker, saturating the fixed volume used for solubility testing and, thus, detrimentally affecting the CBZ solubility [32,33]. Further to this, an increase in viscosity of the solutions with an increase in MAL concentration was observed, which could have also affected the solubility of CBZ [34]..”
Reviewer 2 Report
Comments and Suggestions for Authors
The work proposes solid dispersions with three well-known excipients to increase the dissolution rate of carbamazepine, a poorly soluble drug.
The increase in solubility or dissolution rate through the described method has been known for a long time, as reported in the bibliography. Unfortunately, this study does not provide any novelty compared to the scientific/technological state of the art already acquired. It appears more a report of formulation/pharmaceutical development rather than a research work.
Particular remarks:
An increase in the dissolution rate can be explained, but how can the authors justify an increase in solubility being an intrinsic characteristic of the molecule?
The graphs of the dissolution results are not understandable because the colors of the the curves are too difficult to identify.
Additional experiments are needed to ensure greater sensitivity to the tests. Particularly from a scientific viewpoint, the dissolution tests should be conducted in biorelevant fluids but without any surfactant (sodium lauryl suphate). The addition of a surfactant is needed for Quality Control and to release the production batches: it is essential in pharmaceutical development, but from a scientific viewpoint, it would be better to test the products in its absence.
To compare the drug release profiles, it is not correct to use the student t-test; the f2 similarity test must be used.
Liu, J. P.; Ma, M. C.; Chow, S. C. Statistical evaluation of similarity factor f2 as a criterion for assessment of similarity between dissolution profiles. Drug information journal 1997, 31, 1255-1271
Author Response
Dear reviewer, thank you for taking the time to review this manuscript the comments. Please see the point-by-point response.
Comment 1:
An increase in the dissolution rate can be explained, but how can the authors justify an increase in solubility being an intrinsic characteristic of the molecule?
Response: Thank you very much for the important comment. We respectfully would like to draw your attention to the explanation provided in the manuscript (Lines 595 to 603). These hypotheses were supported by appropriate literature.
“The solubility enhancement among the PMs and SDs can be attributed to the enhanced wettability of the drug particles facilitated by the hydrophilic sugar carriers and the reduced particle size [35,36]. The partial miscibility observed between CBZ and XYL from DSC studies could explain the significant increase in solubility for the CBZ-XYL SDs (p < 0.01) over SDs prepared with MAL and SOR. Similar results were reported in the literature, where predictions made using in silico molecular dynamic simulation correlated with the experimental results of drug carrier miscibility effect on improving the amorphous nature of the SDs and improved solubility [37].”
Comment 2:
The graphs of the dissolution results are not understandable because the colors of the curves are too difficult to identify.
Response: Thank you for highlighting this deficiency. The colours of the graph are now adjusted for easy identification (Figure 15 a,b and c).
Comment 3:
Additional experiments are needed to ensure greater sensitivity to the tests. Particularly from a scientific viewpoint, the dissolution tests should be conducted in biorelevant fluids but without any surfactant (sodium lauryl suphate). The addition of a surfactant is needed for Quality Control and to release the production batches: it is essential in pharmaceutical development, but from a scientific viewpoint, it would be better to test the products in its absence.
Response: Thank you for the valuable comment and we agree with the suggestion. The current study focus was mainly on evaluating the solubility and in-vitro drug release in aqueous media, with the intention to use the results as a basis for future studies to develop a drug delivery system, which will be subjected to dissolution in biorelevant fluids. Furthermore, the dissolution testing was conducted for all samples using the exact same experimental parameters, thus enabling direct comparison and therefore still considered scientifically sound. We are of the opinion that for the exploratory purpose of this study the drug release experimental setup was suitable and do fully agree that for further dosage form development studies dissolution in biorelevant fluids would be critical.
Comment 4:
To compare the drug release profiles, it is not correct to use the student t-test; the f2 similarity test must be used.
Response: Thank you very much for the valuable comment. We have used MDT approach to quantify the average release time and with the intention to have a better comparison between the polyols.
Reviewer 3 Report
Comments and Suggestions for Authors
The manuscript entitled "An investigation into the effect of maltitol, sorbitol and xylitol 2 on the formation of carbamazepine solid dispersions through 3 thermal processing" describes a formulation study of carbamazepine as a solid dispersion in three different sugar carriers, aiming at improving carbamazepine´s solubility. While there is previous research on improving Carbamazepine (CBZ) solubility by solid dispersion formulation, including in sugar carrier, the three polyols investigated in this report do not appear in previous reports. While the investigation is not ground breaking, it increments knowledge on the formulation possibilities of CBZ.
While I consider the methods descriptions complete and clear, and results very well discussed, I do have a major comment on the study design. CBZ was processed by a thermal fusion method, always in a mixture with one of the excipients in different molar ratios. A control of CBZ subject to the same treatment is missing. Interpretation of physico-chemical properties of the solid (sugar) dispersions would be much better substantiated if they were compared to CBZ alone subject to the same treatment.
Minor comments:
Section 2.2.2 Preparation of solid dispersions. "molten product was subsequently cooled to room temperature (RT)." Can You please describe if the products were let cool down, or was the cooling accelerated by any means?
2.2.4 Determination of drug content. The described method relies solely on CBZ extraction by methanol from the solid dispersion. Like this, the obtained assay is a sum of CBZ in a solid dispersion and any "unreacted" of free CBZ. Could the extraction of CBZ from the obtained dispersions have an impact on the obtained solubility? Please elaborate.
Figure 15, the first graph, contains 7 plots but only 6 items in the legend.
Thank You.
Author Response
Dear reviewer, thank you for taking the time to review this manuscript the comments. Please see the point-by-point response.
Comment 1:
While I consider the methods descriptions complete and clear, and results very well discussed, I do have a major comment on the study design. CBZ was processed by a thermal fusion method, always in a mixture with one of the excipients in different molar ratios. A control of CBZ subject to the same treatment is missing. Interpretation of physico-chemical properties of the solid (sugar) dispersions would be much better substantiated if they were compared to CBZ alone subject to the same treatment.
Response: Thank you very much for the valuable comment. A control of CBZ was performed using the fusion method and no significant changes were found for the thermal events. However, this was not reported in the original manuscript considering the standard practice and lack of differences between treated and untreated samples. Similar results were reported in literature. Hence, the current version is amended to indicate that the treated CBZ remain unchanged, supported by literature. Please see Lines 212 - 222.
Comment 2:
Section 2.2.2 Preparation of solid dispersions. "molten product was subsequently cooled to room temperature (RT)." Can You please describe if the products were let cool down, or was the cooling accelerated by any means?
Response: Thank you very much for the comment. This is indicated in the subsequent sentence. “RT cooling was performed by simply leaving the molten product at room temperature (25 °C) until it was solidified.” (Lines 110 – 111).
Comment 3:
2.2.4 Determination of drug content. The described method relies solely on CBZ extraction by methanol from the solid dispersion. Like this, the obtained assay is a sum of CBZ in a solid dispersion and any "unreacted" of free CBZ. Could the extraction of CBZ from the obtained dispersions have an impact on the obtained solubility? Please elaborate.
Response: Thank you very much for the valuable comment. The method was updated to include the quantity of methanol used. . The method was adopted from existing literature, which is now referred.
Comment 4:
Figure 15, the first graph, contains 7 plots but only 6 items in the legend.
Response: Thank you very much for the comment. This is now addressed. The legend is updated to include all samples.
Reviewer 4 Report
Comments and Suggestions for Authors
15.01.2025
A review to evaluate its suitability for publication Type of manuscript:
Article
Title: An investigation into the effect of maltitol, sorbitol and xylitol on the formation of carbamazepine solid dispersions through thermal processing
Authors: Madan Sai Poka, Marnus Milne, Anita Wessels and Marique Aucamр
The peer-reviewed Manuscript (pharmaceutics-3440823) addresses an important challenge for pharmacy in the development of technologies that improve API dissolution and enhance their pharmacokinetic characteristics. In this case, it concerns the use of polyol group excipients - maltitol, sorbitol and xylitol for the manufacture of solid dispersions with the active substance carbamazepine of the antiepileptic class.
The importance of this study to practical pharmacy predetermines careful attention to the results provided, namely:
1. Line 17-19: The authors suggest that there is insufficient research on solid dispersions with polyol inclusions. However, an examination of existing publications on polyols shows that there are a sufficient number of published articles. Therefore, the authors should particularly emphasize the existing gaps in this topic and the lack of necessary practical results, indicating the relevance in this manuscript.
2. It is known about wide application of polyols, in particular, maltite in food industry as an additive sweetener E965, used in dietary food instead of sugar.
However, the use of maltitol is associated with nutritional disorders such as flatulence and diarrhoea. Therefore, authors claiming the safety of this ingredient should consider this fact when recommending this polyol and similar polyols as excipients in the pharmaceutical industry.
3. In the Materials and Methods section, authors should provide labelling of the ingredients used, such as purity, active ingredient content and others.
4. Line 107: «…well-known fusion method….» References should be made to this well-known approach.
5. Line 311: PXRD results for pure carbamazepine require comparison with a diffractogram of a standard from a database such as Powder Diffraction File Seach
6. The authors may need to slightly modify the structure of the results presented, which overlap in the methods of analysis presented from section 3.1 to 3.3. - the same order.
7. Perhaps the results should be structured by method but not by the nature of the solid dispersions?
Respectfully, reviewer
Author Response
Dear reviewer, thank you for taking the time to review this manuscript the comments. Please see the point-by-point response.
Comment 1:
Line 17-19: The authors suggest that there is insufficient research on solid dispersions with polyol inclusions. However, an examination of existing publications on polyols shows that there are a sufficient number of published articles. Therefore, the authors should particularly emphasize the existing gaps in this topic and the lack of necessary practical results, indicating the relevance in this manuscript.
Response: Thank you very much for the valuable comment and we agree with this. The problem statement is revised to clarify it further. Please see below changes made (Lines 17 - 23).
“Polyols are generally regarded as safe (GRAS) and among the approved polyols for market use, maltitol (MAL), xylitol (XYL) and sorbitol (SOR) are among the approved polyols for market use. While xylitol (XYL) and sorbitol, have shown promise in improving the solubility and dissolution rates of poorly soluble drugs, their full potential in the context of improving the solubility of carbamazepine have not been thoroughly investigated. To the best of our knowledge, maltitol (MAL) was not studied previously as a carrier for preparing SDs.”
Comment 2:
It is known about wide application of polyols, in particular, maltite in food industry as an additive sweetener E965, used in dietary food instead of sugar. However, the use of maltitol is associated with nutritional disorders such as flatulence and diarrhoea. Therefore, authors claiming the safety of this ingredient should consider this fact when recommending this polyol and similar polyols as excipients in the pharmaceutical industry.
Response: Thank you very much for the valuable comment and we agree with the associated nutritional drawbacks. However, we would like to highlight that the maximum allowed daily limit for maltitol is 15 g in children and 40 g in adults (Please see the below reference). We also wish to bring to your attention the existing use of polyols, including maltitol in the pharmaceutical industry and established safety limits for human consumption. (Please see Lines 63 to 68).
Saraiva A, Carrascosa C, Raheem D, Ramos F, Raposo A. Maltitol: Analytical Determination Methods, Applications in the Food Industry, Metabolism and Health Impacts. Int J Environ Res Public Health. 2020 Jul 20;17(14):5227. doi: 10.3390/ijerph17145227. PMID: 32698373; PMCID: PMC7400077.
Comment 3:
In the Materials and Methods section, authors should provide labelling of the ingredients used, such as purity, active ingredient content and others.
Response: Thank you very much for the comment. This is now addressed. See Lines 88 to 90.
Comment 4:
Line 107: «…well-known fusion method….» References should be made to this well-known approach.
Response: Thank you very much for the comment. Reference is now included.
Comment 5:
Line 311: PXRD results for pure carbamazepine require comparison with a diffractogram of a standard from a database such as Powder Diffraction File Seach
Response: Thank you very much for the comment. Unfortunately, the institution do not have access to ICDD at this point in time to provide comparison. However, we have compared the diffractogram with two of the published articles (See line 321) to confirm the identification.
Comment 6 & 7:
The authors may need to slightly modify the structure of the results presented, which overlap in the methods of analysis presented from section 3.1 to 3.3. - the same order.
Perhaps the results should be structured by method but not by the nature of the solid dispersions?
Response: Thank you very much once again for the comment. Unfortunately, we could not attend to this as some of the data such as HSM data was not flowing very well, if discussed separately. We respectfully indicate that, for the reasons of logical flow of the information, this comment could not be addressed.
Reviewer 5 Report
Comments and Suggestions for Authors
1. Section 3.1- results and discussion - lines 322-326- why there is a significant difference between DSC and XRD patterns of CBZ-MAL SDs ? The DSC studies (Fig 2) shows the presence of crystallinity though the intensity is reduced in SDs whereas the PXRD (Fig 4) shows a typical amorphous halo. The results are contradicting each other. Is this a technical error in representing the data? How can you explain it? Include explanation in the manuscript and correct it if it is an error from technical point of view
2.Section 3.2- results and discussion - lines 439-443- elaborate the discussion on why the CBZ crystal growth is observed? Include appropriate literature support
3. Section 3.3- results and discussion - lines 530-532 -it was mentioned to observe 2 hallow humps for CBZ-SOR and CBZ-MAL. However in figure 4 PXRD pattern of CBZ_MAL looks like a soft linear curve? How can you explain it? Make the necessary changes in the manuscript and if it is an issue of magnification of curve represent it in such a way that the readers can observe it.
4. Section 3.4- results and discussion - lines 567-578- attribute the results of solubility studies to the chemical nature/ structure of respective polyols and improve the discussion with apt literature support.
5. Section 3.5- results and discussion - In-vitro drug release studies-Include the equation for calculation of MDT in the manuscript.
6. Section 3.5- results and discussion - In-vitro drug release studies-lines -629-632-Corrcet the statement "fusion process reduced the particle size" Fusion is a thermal process and based on this heat assisted process depending on the interaction between drug and polyol the structural changes in crystal occur. It also depend on the process temperature (210C) at which the fusion was carried out. Fusion is different process compared to the physical methods used to reduce the particle size. So write the discussion with improved clarity and explanation.
Author Response
Dear reviewer, thank you for taking the time to review this manuscript the comments. Please see the point-by-point response.
Comment 1:
Section 3.1- results and discussion - lines 322-326- why there is a significant difference between DSC and XRD patterns of CBZ-MAL SDs? The DSC studies (Fig 2) shows the presence of crystallinity though the intensity is reduced in SDs whereas the PXRD (Fig 4) shows a typical amorphous halo. The results are contradicting each other. Is this a technical error in representing the data? How can you explain it? Include explanation in the manuscript and correct it if it is an error from technical point of view.
Response: Thank you very much for the valuable comment. The following explanation (Lines 258 to 269) is presented to explain the phenomenon with appropriate literature support.
“Results obtained for the SDs are in good corelation with the published investigations using the same preparation method [9,10]. This can be further confirmed by calculating the ΔH-values of each phase transition. It was reported that, the ΔH-value associated with the first two events (melting of Form III, recrystallisation to Form I) accounts for approximately 12 J/g and the third event (melting of Form I) accounts for 100 J/g [9]. The experimental values obtained are recorded as 15 J/g and 91 J/g, which closely corresponds with that reported previously. These findings indicate that during the fusion method CBZ may have recrystallised as Form III into a possible amorphous MAL matrix followed by recrystallisation of Form III to Form I during heating. The thermal analysis indicated that the SDs formed could potentially exist as two-phase glass suspensions with CBZ dispersed as fine crystalline particles.”
Comment 2:
Section 3.2- results and discussion - lines 439-443- elaborate the discussion on why the CBZ crystal growth is observed? Include appropriate literature support.
Response: Thank you this valuable comment is noted. The following explanation (Lines 453-457) is presented to explain the phenomenon with appropriate literature support.
“This can be attributed to the fact that the CBZ is dispersed as fine crystalline particles in amorphous SOR, the thermal behaviour of polymorphic drugs with change in temperature could result in high crystallization tendency [20,21]”
Comment 3:
Section 3.3- results and discussion - lines 530-532 -it was mentioned to observe 2 hallow humps for CBZ-SOR and CBZ-MAL. However in figure 4 PXRD pattern of CBZ_MAL looks like a soft linear curve? How can you explain it? Make the necessary changes in the manuscript and if it is an issue of magnification of curve represent it in such a way that the readers can observe it.
Response: Thank you for the comment. It is a technical aspect with the lack of proper magnification. This is now addressed.
Comment 4:
Section 3.4- results and discussion - lines 567-578- attribute the results of solubility studies to the chemical nature/ structure of respective polyols and improve the discussion with apt literature support.
Response: Thank you very much for the highly valuable comment. This has been now addressed. Please see below text. (Line 605 - 615).
“The physicochemical properties, such as number of hydroxyl (-OH) groups along its carbon chain, play a central role in solubility by forming hydrogen bonds with drug molecules. However, the structure of these polyols can enhance or reduce solubility depending on factors like molecular size and molecular weight, where larger molecules may face steric hindrance and experience less efficient interaction with solvent molecules. In addition polyols that are linear or acyclic tend to have greater solubility in water over the cyclic polyols [38,39]. This can further explain the high solubility of CBZ in the presence of xylitol and sorbitol over maltitol. Despite maltitol possessing a large number of hydroxyl groups (9), the cyclic nature of the compound and its high molecular weight (344.31 g/mol) seems to have a negative effect on the solubility.”
Comment 5:
Section 3.5- results and discussion - In-vitro drug release studies-Include the equation for calculation of MDT in the manuscript.
Response: Thank you for the comment. The MDT is calculated using DD Solver software. This is now mentioned in the manuscript (Line 648).
Comment 6:
Section 3.5- results and discussion - In-vitro drug release studies-lines -629-632-Corrcet the statement "fusion process reduced the particle size" Fusion is a thermal process and based on this heat assisted process depending on the interaction between drug and polyol the structural changes in crystal occur. It also depends on the process temperature (210C) at which the fusion was carried out. Fusion is different process compared to the physical methods used to reduce the particle size. So, write the discussion with improved clarity and explanation.
Response: Thank you very much for highlighting this aspect and providing explanation. This is now addressed as following. (Lines 656-657).
“In addition, the fusion process could have resulted in the dispersion of drug at molecular level, availing high surface area for contact with the dissolution medium.”
Round 2
Reviewer 2 Report
Comments and Suggestions for Authors
Unfortunately, the major problem of this article persists: this technique to improve the passage into solution of poorly soluble drugs has been known for at least 70 years, and the authors themselves have already published works on its application. http://dx.doi.org/10.3390/cryst13111568; http://dx.doi.org/10.1016/j.ejps.2019.105057; http://dx.doi.org/10.1208/s12249-015-0302-4 and a review: http://dx.doi.org/10.3390/pharmaceutics15112557
Simply changing the drug and/or excipients does not represent scientific novelty. Mass-produced articles that change only the molecule present should not be considered innovative and could seriously undermine the reputation of the scientific journal.
Responses to the proposed observations: Unfortunately, the authors limited themselves to thanking the reviewer's suggestions, but without solving the problems raised by simply avoiding them.
Point 1.
I beg to differ: for any chemist, wettability and the reduction of particle size or, better, dispersion of drug at the molecular level increases the dissolution rate of a molecule (but not for all of them) and this can be an advantage for many drugs, but it cannot modify the solubility which is a completely different characteristic intrinsic of the molecule itself.
Point 3: To have an advantage in drug absorption, in vitro dissolution tests must be conducted at least in biorelevant fluids (which can simulate in vivo conditions); the results in water can be very different and, therefore, misleading.
Point 4: to evaluate the difference or similarity of a dissolution profile (which is a complex and time-extended phenomenon), you cannot use the Student's t-test to a single point, but the f2 similarity test or possibly a two one-sided t-test (TOST) approach at each dissolution time point that needs a priori definition of similarità concerning the maximum acceptable difference between the two mean dissolution profiles.
Author Response
Dear Reviewer,
Please see the response below to all the comments. we hope that they are sufficiently addressed.
Comment 1: Unfortunately, the major problem of this article persists: this technique to improve the passage into solution of poorly soluble drugs has been known for at least 70 years, and the authors themselves have already published works on its application. http://dx.doi.org/10.3390/cryst13111568; http://dx.doi.org/10.1016/j.ejps.2019.105057; http://dx.doi.org/10.1208/s12249-015-0302-4 and a review: http://dx.doi.org/10.3390/pharmaceutics15112557
Simply changing the drug and/or excipients does not represent scientific novelty. Mass-produced articles that change only the molecule present should not be considered innovative and could seriously undermine the reputation of the scientific journal.
Responses to the proposed observations: Unfortunately, the authors limited themselves to thanking the reviewer's suggestions, but without solving the problems raised by simply avoiding them.
Response: Dear reviewer, we do acknowledge the amount of work published in this area. However, we would like to indicate that one of the polyols studied (maltitol) was not studied before and we consider this as addition of new knowledge to the body of science.
We also regret for not communicating the response effectively in previous submission. In this submission, we tried to bring our submission as clear as possible with appropriate references. We also have amended the sections, where possible.
Point 1.
I beg to differ: for any chemist, wettability and the reduction of particle size or, better, dispersion of drug at the molecular level increases the dissolution rate of a molecule (but not for all of them) and this can be an advantage for many drugs, but it cannot modify the solubility which is a completely different characteristic intrinsic of the molecule itself.
Response:
We do understand the intrinsic aspect of drug solubility. However, we respectfully would like to bring your attention to the effect of particle size on solubility enhancement. Below is the text quoted from literature review and citations are provided for your reference. In supporting the hypothesis behind the solubility enhancement, we used published research that was cited in the manuscript (Line 602-610).
“The solubility of drug is often intrinsically related to drug particle size; as a particle becomes smaller, the surface area to volume ratio increases. The larger surface area allows greater interaction with the solvent which causes an increase in solubility.”
References:
Savjani KT, Gajjar AK, Savjani JK. Drug solubility: importance and enhancement techniques. ISRN Pharm. 2012;2012:195727. doi: 10.5402/2012/195727. Epub 2012 Jul 5. PMID: 22830056; PMCID: PMC3399483.
Ainurofiq, Ahmad & Putro, DavidSarono & Ramadhani, DheaAqila & Putra, GemalaMahendra & Santo, LauraDa. (2021). A review on solubility enhancement methods for poorly water-soluble drugs. Journal of Reports in Pharmaceutical Sciences. 10. 137. 10.4103/jrptps.JRPTPS_134_19.
Point 2: To have an advantage in drug absorption, in vitro dissolution tests must be conducted at least in biorelevant fluids (which can simulate in vivo conditions); the results in water can be very different and, therefore, misleading.
Response:
As indicated in the previous submission, we do agree with your comment. The study was designed to explore the solubility and dissolution enhancement of the drug in the presence of polyols (both the physical mixtures and solid dispersions). Hence, was used only aqueous media. However, these results will be used as a basis for future studies to develop a drug delivery system, which will be subjected to dissolution in biorelevant fluids.
Point 3: to evaluate the difference or similarity of a dissolution profile (which is a complex and time-extended phenomenon), you cannot use the Student's t-test to a single point, but the f2 similarity test or possibly a two one-sided t-test (TOST) approach at each dissolution time point that needs a priori definition of similarity concerning the maximum acceptable difference between the two mean dissolution profiles.
Response:
We understand your comment with regards to using the similarity factor for comparing the dissolution profiles. We have now included this data (Table 2) along with other model independent parameters (MDT and %DE). In line with the addition, sub-section 2.2.7 is added under methods). The updated discussion is now between the lines 654 to 682. We hope the data and discussion provided would address the comment.
Round 3
Reviewer 2 Report
Comments and Suggestions for Authors
I appreciate the authors' effort, and I am sorry to have to reiterate that the fundamental problem remains: frankly, the simple application of a technique that has been well-known for years cannot be considered a scientific innovation. The same applies to the choice of the best drug-excipient combination. Otherwise, the daily work of every good pharmaceutical formulator should always be published in a scientific journal.